# Dielectric Spectroscopy Using Dual Reflection Analysis of TDR Signals [note 1]

**DOI:** 10.3390/s19061299

**Published:** 2019-03-14

**Authors:** Yin Jeh Ngui, Chih-Ping Lin, Tsai-Jung Wu

**Affiliations:** Department of Civil Engineering, National Chiao Tung University, 1001, Ta-Hsueh Road, Hsinchu City 300, Taiwan; astronomer6573@gmail.com (Y.J.N.); tsaijung.cv00@g2.nctu.edu.tw (T.-J.W.)

**Keywords:** time-domain reflectometry (TDR), dielectric spectroscopy, dual reflection analysis

## Abstract

Time-domain reflectometry (TDR) has been a powerful tool for measuring soil dielectric properties. Initiating from apparent dielectric constant ($K_{a}$) measurement up until apparent and complex dielectric spectroscopies, the embedded information in the TDR signal can be extracted to inspire our understanding of the underlying dielectric behaviors. Multiple full waveform inversion techniques have been developed to extract complex dielectric permittivity (CDP) spectrum, but most of them involved prior knowledge of input function and tedious calibration. This rendered the field dielectric spectroscopy challenging and expensive to conduct. Dual reflection analysis (DRA) is proposed in this study to measure CDP spectrum from 10 MHz to 1 GHz. DRA is a simple, robust, model-free, and source-function free algorithm which requires minimal calibration effort. The theoretical framework of DRA is established and the necessary signal processing procedures are elaborated in this study. Eight materials with different dielectric characteristics are selected to evaluate DRA’s performance, by using both simulated and experimental signals. DRA is capable of measuring non-dispersive materials very well, whereas dispersive materials require the assistance of a long-time-window (LTW) extraction method to further extend the effective bandwidth. The DRA approach is suitable for field applications that can only record a limited amount of data points and in-situ dielectric spectroscopy.

## 1. Introduction

Time-domain reflectometry (TDR) has been a robust tool for measuring soil dielectric properties, ranging from apparent dielectric constant ($K_{a}$) measurement to both apparent and complex dielectric spectroscopies. TDR has since provided accurate complex dielectric permittivity (CDP) estimation in soil sciences [1,2,3,4] throughout the MHz to GHz frequency bands. Originally, implementation of TDR in soil water content relied upon the empirical correlation between volumetric water content and $K_{a}$ [5]. The $K_{a}$ is typically estimated by analyzing time domain TDR signals with the tangent line method and relies on major reflections from the head and end of probe sensing section. However, the corresponding effective frequency band of the measured $K_{a}$ is unclear, leading to potential dependency on electrical conductivity, dielectric loss, measurement system configurations, and other factors [6,7,8]. Measurement consistency and accuracy of data interpretation using tangent line methods may also vary due to the practice of different data handlers and signal qualities. Lin et al. [9] proposed a phase velocity analysis (PVA) method to extract the apparent dielectric permittivity (ADP) spectrum from TDR signals, introducing an apparent dielectric constant with frequency information from 10 MHz to 1 GHz. Time-domain signals of the probe head and the probe end are truncated and differentiated into independent pulses. PVA measures the ADP spectrum by explicitly solving for the phase angle difference in the frequency domain after applying a fast Fourier Transform (FFT) to the time domain signals. PVA is shown to be capable of accurately determining the ADP of various liquids and soils without any prior knowledge of system input function or leading transmission line setup [4,10]. Although PVA is a model-free and inversion-free algorithm, both real and imaginary parts of CDP are yet to be decoupled in order to identify dielectric polarization behavior and energy dissipation characteristics due to electrical conductivity and dielectric loss.

To extend the measurement capability of TDR, various literatures investigated the frequency domain analysis approach of TDR signals, involving the inversion of the complex dielectric permittivity (CDP) spectrum from input functions and scattering functions [2,3,11,12,13]. Relaxation models were used in these inversions to characterize the material under test (MUT) with optimized model parameters (Debye or Cole–Cole). This frequency domain approach is highly capable of measuring the CDP spectrum within a laboratory environment and a short leading cable. Since the input function and calibration of leading mismatched sections is required as a priori in CDP inversion, field dielectric spectroscopy would become challenging as the measurement setup and calibration are tedious and prone to in-situ disturbances. In-situ dielectric spectroscopy may reveal more embedded dielectric information pertinent to soil moisture dynamics and soil contamination monitoring.

A vector network analyzer (VNA) is widely used to characterize the CDP spectrum of the MUT in a laboratory environment [14,15]. Measurements from a VNA are very accurate and reliable, provided that the system calibration is handled properly. A VNA requires a long calibration time and demands specifically designed sample holders and short cables. If the VNA measurement system is moved or restarted, system recalibration is often needed as the interface connections may slightly displace during the instrument repositioning. This may render field dielectric spectroscopy and monitoring extremely difficult and financially infeasible, as in-situ environments are typically harsher than laboratories.

Lin et al. [16], therefore, proposed a robust and model-free multiple reflection analysis (MRA) of TDR signals to measure the field CDP spectrum. The MRA approach decomposes the first main reflection and the subsequent multiple reflections from sensing sections. Their spectral ratios were compared and fitted with model-free inversion to generate CDP spectrums. Nevertheless, multiple reflections may not be fully acquired up to the steady state condition, which may be attributable to the instrumentation limitation and the necessity to seek balance between time resolution and data point amounts. In light of this, this study proposed and evaluated a dual reflection analysis (DRA) approach on TDR signals to measure both real and imaginary parts of the CDP spectrum, from 10 MHz to 1 GHz, utilizing a more typical recording time window that does not include higher order multiples. The theoretical framework of DRA is presented first, followed by the experimental setup to generate laboratory TDR signals. DRA is examined next, using both simulated and experimental signals of various MUTs, where the resulting CDP spectrum and measurement findings in this study are discussed.

## 2. Materials and Methods

### 2.1. Theoretical Framework of DRA

In general, a TDR instrument sends a step pulse input into the transmission line, up until the end of the probe sensing section. The propagating electromagnetic (EM) wave interacts with the particles of the MUT at the sensing section that induces polarization, in which this effect would be reflected on the reflection signal recorded by the oscilloscope of the TDR system. The first two major reflections of interest come from the beginning and end of the probe sensing section, particularly for an open-ended probe with an impedance-matched probe head (to the leading cable). These major reflections are due to discontinuity in characteristic impedance ($Z_{c}$) between the leading cable, the probe sensing section, and the open-ended section. Following the definition in Lin et al. [9], Figure 1 demonstrates the propagation of the input signal X, propagating along the leading section up to the probe head, where the system functions of this forward propagation and the returning backward propagation are denoted as F and B. Interface I is the interface between the probe head and the probe sensing section, in which the first major reflection signal (denoted as $r_{1}$) reflects at this location and its spectral response $R_{1}$ (the lower case and the capital letter represent time domain and frequency domain variables, respectively), which can be expressed as follows:(1)$$R_{1}=X\;\cdot F\cdot \rho_{1}\cdot B,$$
where $\rho_{1}$ is the reflection coefficient at Interface I. The reflection coefficient, at any specific interface, is a function of the characteristic impedance ($Z_{c}$) of the prior and subsequent sections. Therefore, $\rho_{1}$ can be written as follows:(2)$$\rho_{1}=\frac{Z_{cs}-Z_{ch}}{Z_{cs}+Z_{ch}},$$
where $Z_{ch}$ and $Z_{cs}$ are the $Z_{c}$ of the probe head and the probe sensing section. The value $Z_{c}$ can be determined as the ratio of the geometric impedance ($Z_{p}$, also the characteristic impedance in air) to the square root of the material’s CDP. CDP is denoted by $\varepsilon^{*}\left(f\right)$, as follows:(3)$$\varepsilon^{*}\left(f\right)=\varepsilon^{\prime }\left(f\right)+j\varepsilon \prime \prime \left(f\right),$$
where f is the frequency, and $\varepsilon^{\prime }$ and ε″ are real parts and the imaginary parts of the frequency-dependent CDP. $\rho_{1}$ can be further arranged into the following:(4)$$\rho_{1}=\frac{1-\frac{Z_{ch}}{Z_{ps}}\sqrt{\varepsilon^{*}\left(f\right)}}{1+\frac{Z_{ch}}{Z_{ps}}\sqrt{\varepsilon^{*}\left(f\right)}},$$
where $Z_{ch}$ and $Z_{ps}$ are the characteristic impedance of the probe head (leading cable) and the geometric impedance of the probe sensing section, respectively.

Apart from reflection $r_{1}$, part of the EM wave propagates into the sensing section and the EM wave is exposed to the MUT contained within the sensing region. Carrying the polarization influence from surrounding material, this EM wave reaches the end of the open-ended coaxial probe (Interface II), where the second main reflection, $r_{2}$, occurred. For an open-ended interface, $\rho_{2}$ is equal to 1 and the EM wave travels back from Interface II to Interface I again. Following the described wave propagation path, the spectral response of $r_{2}$ is expressed as the following:(5)$$R_{2}=X\;\cdot F\cdot (1+\rho_{1})\cdot H\left(2L,\;\varepsilon^{*}\left(f\right)\right)\cdot \left(1-\rho_{1}\right)\cdot B,$$
where L is the probe sensing length. The value H is the system function of the sensing section that accounts for the phase change and the attenuation of the EM wave during its transmission, which is denoted as follows:(6)$$H\left(x,\;\varepsilon^{*}\left(f\right)\right)=\exp\left[-\gamma \left(f\right)x\right],$$
where x is the propagation distance (x=2L when the EM wave traveled back and forth along the probe), whereas γ is the propagation constant, as follows:(7)$$\gamma \left(f\right)=\frac{j2\pi f}{c}\sqrt{\varepsilon^{*}\left(f\right)},$$
where *j* is $\sqrt{-1}$, and c is the speed of light.

The DRA ratio can now be computed by comparing the spectral ratios of $r_{1}$ and $r_{2}$ from Equations (1) and (5), derived as follows:(8)$$DRA=\frac{R_{2}}{R_{1}}=\frac{{\left(1-\rho_{1}\right)}^{2}}{\rho_{1}}\cdot H\left(2L,\;\varepsilon^{*}\left(f\right)\right).$$
The DRA ratio above demonstrates that the input function forward and backward propagation system functions are cancelled out, indicating that this approach is independent of the input function.

### 2.2. Signal Processing and Parameter Calibration for DRA Implementation

Prior to DRA matching for CDP measurement, signal processing is required to properly extract the necessary waveforms from time-domain signals. As observed in Equation (8), $R_{1}$ and $R_{2}$ are the required spectral components of time-domain components $r_{1}$ and $r_{2}$. Figure 2a shows a typical recorded TDR step signal when the sensing probe is fully immersed in water. We can differentiate this step pulse signal into a pulse signal, as shown in Figure 2b. Typically, selecting the time window extraction range for $r_{1}$ and $r_{2}$ is rather straightforward, in which both step and impulse signals can be compared to assist in their time window selection. The value $r_{1}$ starts from the matched leading section until the point before the first reflection arriving from probe end, denoted as $t_{r1a}$ and $t_{r1b}$ respectively in Figure 2a,b. The value $t_{r1b}$ may be selected at the lowest point in the step signal and usually where the differentiated signal passes through the zero point. As for $r_{2}$, the selection range begins from $t_{r1b}$ up to the point before the next multiple reflection, denoted as $t_{r2a}$. For certain dispersive signals, multiple reflections of the differentiated signal may not pass through zero point, so $t_{r2a}$ can be selected at the lowest point prior to the next multiple reflection.

After truncating $r_{1}$ and $r_{2}$, following the aforementioned procedure, zero-padding is performed on both signals to increase the frequency resolution while maintaining the original time resolution. The amount of data points padded prior to and after the truncated signals depends on the acquired time step and the desired frequency step. For example, a TDR signal of 5 ps resolution is padded to 40,000 data points in order to achieve frequency resolution of 5 MHz. A fast Fourier Transform is performed on the padded $r_{1}$ and $r_{2}$ to generate their respective spectral domain, $R_{1}$ and $R_{2}$. For noisy signals, a low-pass filter or a Tukey window can be applied to the extracted signal to prevent spectral leakage during the fast Fourier Transform [16], as shown in Figure 2c,d. The DRA ratio is finally generated from the TDR signal by comparing the spectral ratios as shown in Equation (8).

The CDP spectrum of the MUT is measured by optimizing $\varepsilon^{*}\left(f\right)$ in Equation (8) to fit the measured DRA ratio at each frequency step. Note that the $\varepsilon^{*}\left(f\right)$ of interest is in both $\rho_{1}$ and $H\left(2L,\;\varepsilon^{*}\left(f\right)\right)$. This can be achieved by integrating both the optimization algorithm and the iterated initial guess approach proposed by Lin et al. [16]. The latter approach introduces an initial guess for a higher frequency step, based on the result from a lower frequency step. This is important for CDP inversion at higher frequencies as the cost function structure of DRA at higher frequency ranges tend to form more local minimums than lower frequencies. If an iterated initial guess approach is not implemented in DRA optimization, the provided initial guess may become a critical issue for DRA to obtain the CDP spectrum accurately. Therefore, an arbitrary initial guess is assigned to the lowest frequency to optimize the correct CDP to fit the measured DRA ratio, frequency by frequency, until the whole spectrum is fully generated from 10 MHz to 1 GHz.

As for system parameter calibration in the DRA approach, there are only two system parameters to be calibrated once, the length (L) and geometric impedance ($Z_{p1}$) of the sensing probe. The probe length can be easily calibrated using any MUT with a known CDP spectrum, such as distilled water. For coaxial probes, $Z_{ps}$ can be calculated directly from the geometric impedance equation which follows below [17]:(9)$$Z_{p}=\frac{1}{2\pi }\sqrt{\frac{\mu_{0}}{\varepsilon_{0}}}\ln\frac{D}{d},$$
where $\mu_{0}$ is the vacuum permeability, $\varepsilon_{0}$ is the vacuum permittivity, and *D* is the inner diameter of the outer conductor while *d* is the outer diameter of the inner conductor. For sensing probes with irregular cross sections, $Z_{ps}$ can also be calibrated using a calibration material with a known CDP. Compared to other methods that are capable of complex dielectric spectroscopy, system parameters to be calibrated in DRA are way less and the calibration procedure simply requires one known material. The DRA approach only requires system calibration once after the sensing section is assembled, without the need of supplying short, open, and load (SOL) conditions during the calibration. The broadband TDR reflectometer may also be replaced without recalibrating the DRA’s system parameters, as the difference of source function is cancelled out in the DRA ratio. Other instruments, such as a VNA, require significantly longer system calibration time and require calibration whenever the measurement system is moved or restarted.

### 2.3. Numerical Simulation Parameters

Simulated TDR signals were generated synthetically to compare the DRA results against the experimental signals, in order to investigate the potential difference between simulated and experimental signals. A transmission line model, consisting of three sections, was formulated to emulate experimental results, including a 42 m coaxial cable with a characteristic impedance of 50 Ω, an impedance-matched 50 Ω probe head 10 cm in length, and a 17.2 cm sensing section with a geometrical impedance of 97 Ω. Artificial TDR signals were simulated with the wave propagation model from Lin and Tang [18] and using the Cole–Cole function [19] as the dielectric model of the simulated MUT. The Cole–Cole function characterizes materials’ dielectric behavior using five parameters, denoted as follows:(10)$$\varepsilon^{*}\left(f\right)=\varepsilon_{\infty }+\frac{\varepsilon_{dc}-\varepsilon_{\infty }}{1+{\left(\frac{jf}{f_{rel}}\right)}^{1-\beta }}-\frac{j\sigma }{2\pi f\varepsilon_{0}},$$
where $\varepsilon_{\infty }$ and $\varepsilon_{dc}$ are the dielectric constant at infinite and direct current (0 Hz) frequencies, respectively, $f_{rel}$ is the relaxation frequency in Hz, β is the spectral symmetrical shape parameter of dielectric loss, and σ is the electrical conductivity. The slight cable resistance effect was adopted in the forward simulation and the cable resistance loss factor ($\alpha_{r}$) was set as 50, which is almost equivalent to a P3-500 coaxial cable. Eight MUT with different dispersion characteristics and dielectric constants were selected in this study for DRA evaluation, namely distilled water, tap water, acetone, air, methanol, ethanol, isopropanol, and butanol. The TDR signals of the first four MUT are relatively non-dispersive compared to the latter four alcohol type MUT. Distilled water and tap water were selected to investigate the difference in electrical conductivity, while air was selected to evaluate the performance of DRA at very low dielectric constant. Various alcohols were opted to assess the DRA against different dispersion degrees. The Cole–Cole parameters of the eight MUT are tabulated in Table 1 and are used as the dielectric models in the subsequent signal simulation.

### 2.4. Experimental Setup

A laboratory measurement system is used to evaluate the performance of DRA in eight MUT with different dispersion degrees and dielectric characteristics, as shown in Figure 3. This measurement system is comprised of a broadband TDR device, a 42 m 50 Ω coaxial leading cable, a 10 cm 50 Ω-impedance matched modularized coaxial probe head, and a 17.2 cm coaxial sensing section with a geometric impedance of 97 Ω. The 42 m long cable is not ideal for a laboratory setup but was used on purpose to evaluate the applicability of the proposed approach in a field setup. The broadband TDR device used in this study is a commercial TDR3000 device from Sympuls Aachen (Aachen, Germany) with 3 GHz bandwidth and a risetime of approximately 97 ps. TDR signal acquisition is not limited to this specific pulsing device and any readily available TDR device can be used to measure the experimental signals. Time-domain measurement is performed by utilizing the TDR3000 to inject an EM step signal into the coaxial transmission line system and allowing the EM wave to be exposed to the MUT within the coaxial sensing section. All time-domain TDR signals were acquired with 5 ps sampling time and 10,000 data points, in a controlled room temperature of 20 °C (±0.2 °C), starting from approximately 1 m before the matched probe head.

## 3. Results and Discussion

### 3.1. DRA in Non-Dispersive MUTs

Simulated and experimental signals of the four non-dispersive MUT are shown in Figure 4a,b, respectively. The noise degree of the experimental signal is higher than the simulated signal due to ambient noise, in which this study has reduced the influence by waveform stacking using 12 sets of data. These time-domain TDR signals are processed according to Section 2.3 using a fast Fourier Transform to generate their measured DRA ratio. The DRA ratio of simulated and experimental signals are marked in darker and lighter colors in Figure 5c. Only the DRA ratio of distilled water is shown in Figure 5c for presentation conciseness. In general, the noise level in the experimental signal was much higher than the simulated signal, leading to a higher oscillation error in the DRA ratio. The induced noise caused the oscillation error of the optimized CDP spectrum to be relatively significant, as shown, using distilled water as an example.

The measured CDP spectrum of the four MUT are shown in Figure 5, Figure 6, Figure 7 and Figure 8, where results from both the simulated and experimental signals are stacked together to investigate the effects from a real-world scenario. As presented in the four figures, the DRA is shown to be capable of accurately measuring and decoupling the real and imaginary parts of the CDP spectrum for a certain frequency range, directly from the time-domain signals. Even the resulting CDP spectrum for air, which has very low dielectric constant and a short travel time, is shown to correctly follow the theoretical value in Figure 8.

The effective CDP spectrum bandwidth of the experimental signals is slightly narrower than of the simulated signals. The effective bandwidth is influenced by several factors. The long leading cable physically filters out the spectral content of higher frequencies. The existence of random noise aggravates this condition, thus reducing the higher effective frequency region of the experimental signals. As for the lower bound frequency, cable resistance effect increases the risetime of reflected waves and causes tail leakage of $r_{1}$ into the extracted $r_{2}$. This signal leakage mainly affects the lower frequency components, causing the estimation to gradually deviate from the theoretical value below 100 MHz. Similar problems are also observed in PVA and MRA approaches.

### 3.2. DRA in Dispersive MUTs

Apart from non-dispersive MUT, DRA was further evaluated using alcohols with high dispersion, where methanol, ethanol, isopropanol, and butanol were used. Their time-domain signals are presented in Figure 9, in which simulated and experimental signals are shown in subfigure (**a**) and (**b**) respectively. Some experimental signals differed from the simulated signals due to the input scaling consistency of the TDR pulser and no amplitude scaling was applied to the experimental signals. Nevertheless, this is not an issue for DRA as this input scaling effect would be eliminated through the spectral comparison of $r_{1}$ and $r_{2}$.

DRA generated CDP spectrums for the four dispersive MUT are shown from Figure 10, Figure 11, Figure 12 and Figure 13. Similar to the non-dispersive MUT, DRA can measure the CDP spectrum of the four dispersive MUT within a certain effective frequency range. However, no matter if simulated or experimental signals, the effective frequency range for the dispersive MUT were significantly narrower than the non-dispersive MUT. The upper effective frequency bounds of dispersive MUT are located mostly around the peak of imaginary CDP, which is near their corresponding relaxation frequencies. As alcohol type MUT are highly dispersive, truncation of $r_{2}$ could not be selected properly at the zero point, instead the lowest point before the next multiple reflection was selected as $t_{r2a}$. This extraction window selection introduced a significant signal truncation effect, where a sudden drop occurred between $t_{r2a}$ and the padded zeros, as demonstrated in Figure 14b.

### 3.3. Long-Time-Window (LTW) Selection for Dispersive Signals

Since the signal truncation effect has a significant impact on the CDP spectrum of dispersive MUT, the extraction time window of $t_{r2a}$ was investigated for potential improvements to DRA. Figure 14 uses the time-domain signal of isopropanol as an example. A long-time-window (LTW) approach is adopted alongside with DRA to assist in the CDP inversion of dispersive MUT. As mentioned in a previous section, dispersive materials often present a challenge to select a proper $t_{r2a}$, in which the sudden drop between $r_{2}$ and the subsequent zero-padding should be minimized as little as possible. Sudden drops after the selected $t_{r2a}$ for methanol and ethanol are relatively small than isopropanol and butanol, as their relaxation frequency are higher than the latter two.

Using the experimental signal of isopropanol as an example (Figure 14b), the first zero crossing point in the differentiated signal lies around the end of the recorded signal. As opposed to the normal time window (NTW) selection of $r_{2}$, this study attempted a LTW approach on isopropanol that included the multiple reflections after $r_{2}$ into the spectral component of $r_{2}$ ($R_{2}$). The $r_{2}$ value of isopropanol was extracted using NTW and LTW respectively, which were processed with the same DRA procedures to produce the CDP spectrum shown in Figure 15. Although the LTW approach may violate the formulation of $R_{2}$ (i.e., the extracted $r_{2}$ is contaminated by small higher order multiples), this approach is beneficial towards reducing the signal truncation effect, as seen in Figure 15. Some oscillation and deviation presented in the LTW generated spectrum, but the overall effective frequency range was significantly increased compared to the NTW. This is a side effect due to the contamination of multiples in the extracted $r_{2}$. As such, better characterization of highly dispersive MUT can be conducted by the full MRA approach [16], which fully considers all multiple reflections in its formulations but requires longer recording time to allow the TDR signal to fully reach the steady state.

## 4. Conclusions

In this study, Dual reflection analysis (DRA) of TDR signals were proposed and evaluated to perform dielectric spectroscopy from 10 MHz to 1 GHz. The proposed DRA method is a simple, robust, model-free, and source-function free algorithm to measure the complex dielectric permittivity (CDP) spectrum with decoupled real and imaginary parts. DRA extracts the first two major reflections from the sensing section and fast Fourier Transforms the time-domain signal into spectral DRA ratios for CDP spectrum fitting. This approach is validated using eight materials with different dielectric characteristics and its performance is investigated through the comparison of simulated and experimental data. DRA is capable of measuring the CDP spectrum of non-dispersive materials very well, with satisfactory effective frequency bandwidth. As for dispersive alcohol type materials, DRA with long-time-window (LTW) extraction is recommended to suppress the significant signal truncation effect and extend the effective frequency range. The DRA approach is suitable for field applications that can only record a limited amount of data points and in-situ dielectric spectroscopy. This approach is potentially advantageous in geotechnical and geo-environment industries for soil moisture monitoring, groundwater and soil contamination monitoring, liquid quality detection, and so forth. This method can be extended to multiple reflection analysis (MRA), which fully considers all multiple reflections in its formulations but requires longer recording time to allow the TDR signal to fully reach the steady state.

## Figures and Tables

**Figure 1 sensors-19-01299-f001:**
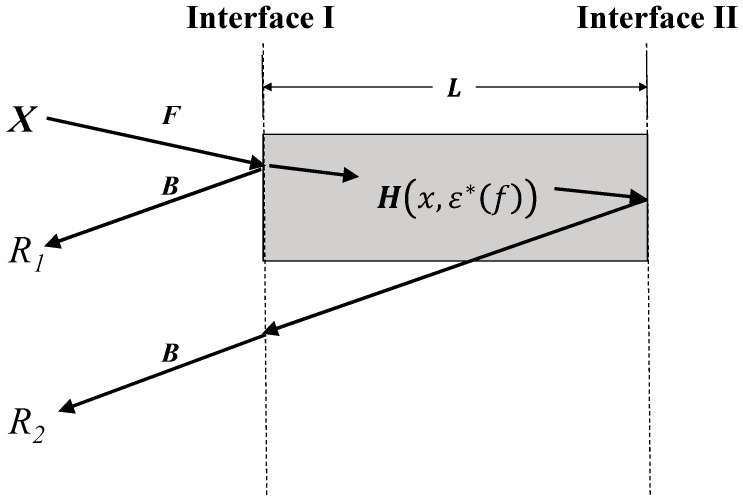
Illustration of the ray tracing diagram for the first two major reflections.

**Figure 2 sensors-19-01299-f002:**
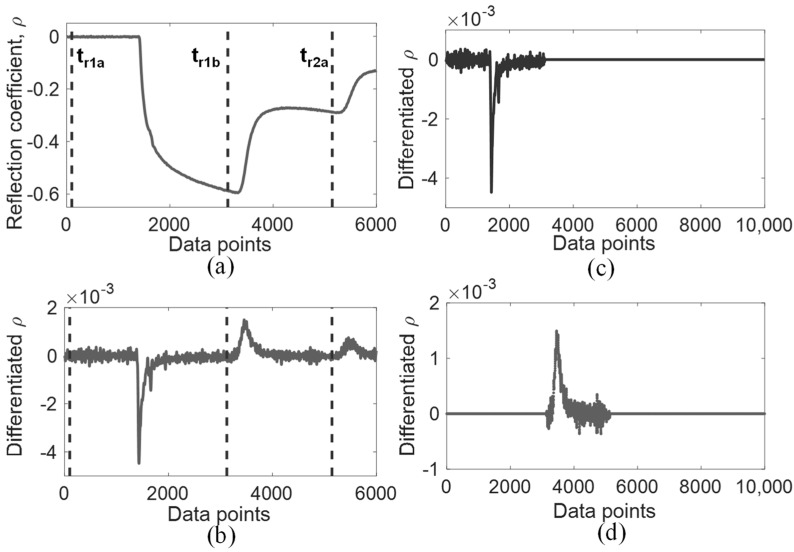
DRA signal processing illustration: (**a**) TDR step signal in tap water; (**b**) Differentiated signal of step signal (**a**); (**c**) Extracted signal $r_{1}$; (**d**) Extracted signal $r_{2}$.

**Figure 3 sensors-19-01299-f003:**
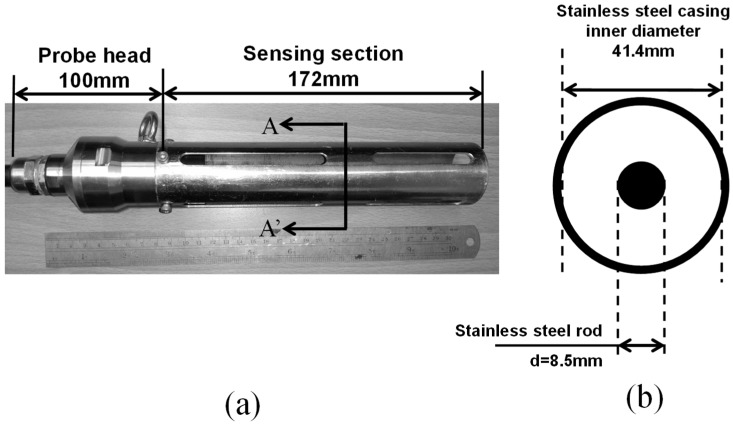
Laboratory measurement setup for DRA. (**a**) Coaxial probe; (**b**) Cross-section of (**a**).

**Figure 4 sensors-19-01299-f004:**
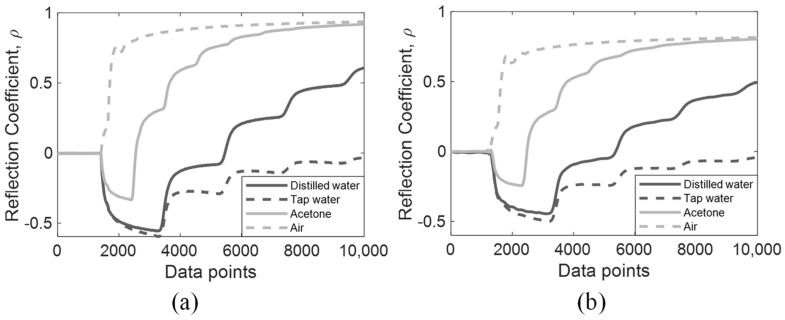
Non-dispersive MUT. (**a**) Simulated signals; (**b**) Experimental signals.

**Figure 5 sensors-19-01299-f005:**
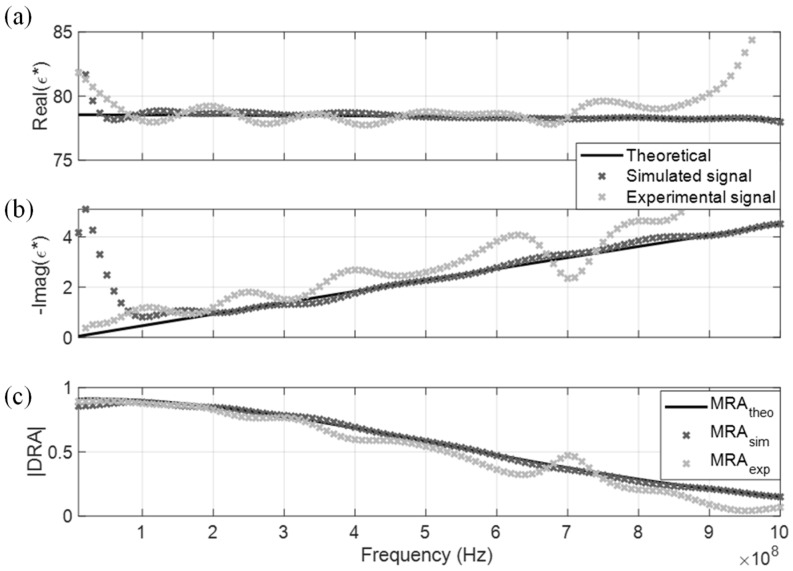
DRA in distilled water. (**a**) Real CDP; (**b**) Imaginary CDP; (**c**) DRA ratio.

**Figure 6 sensors-19-01299-f006:**
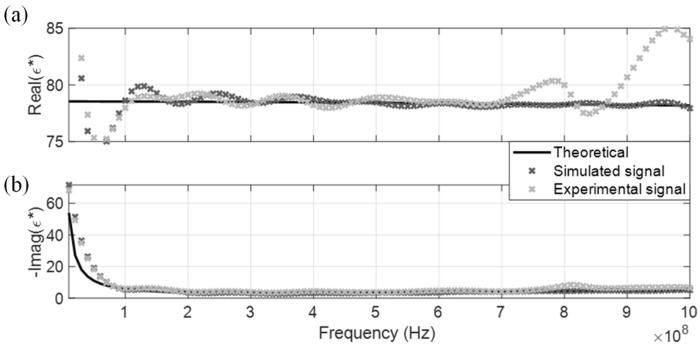
DRA in tap water. (**a**) Real CDP; (**b**) Imaginary CDP.

**Figure 7 sensors-19-01299-f007:**
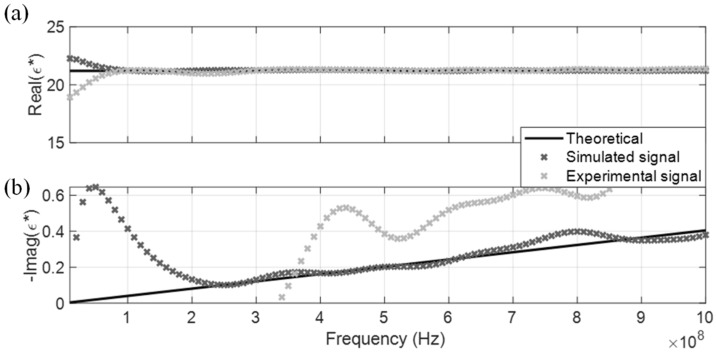
DRA in acetone. (**a**) Real CDP; (**b**) Imaginary CDP.

**Figure 8 sensors-19-01299-f008:**
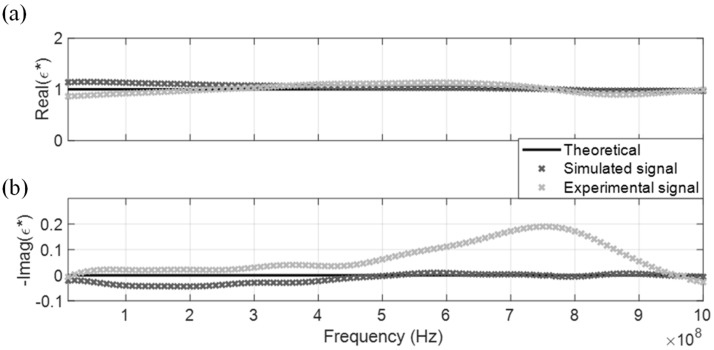
DRA in air. (**a**) Real CDP; (**b**) Imaginary CDP.

**Figure 9 sensors-19-01299-f009:**
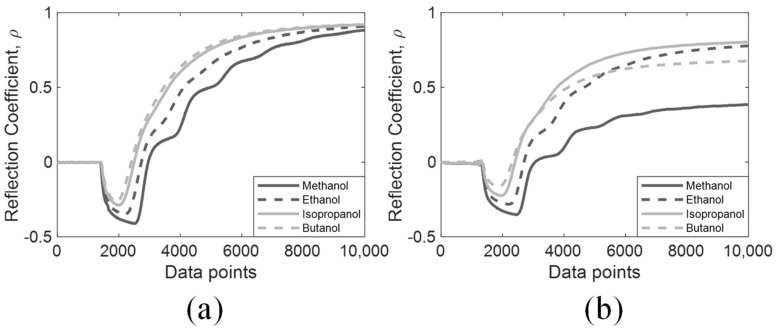
Dispersive MUT. (**a**) Simulated signals; (**b**) Experimental signals.

**Figure 10 sensors-19-01299-f010:**
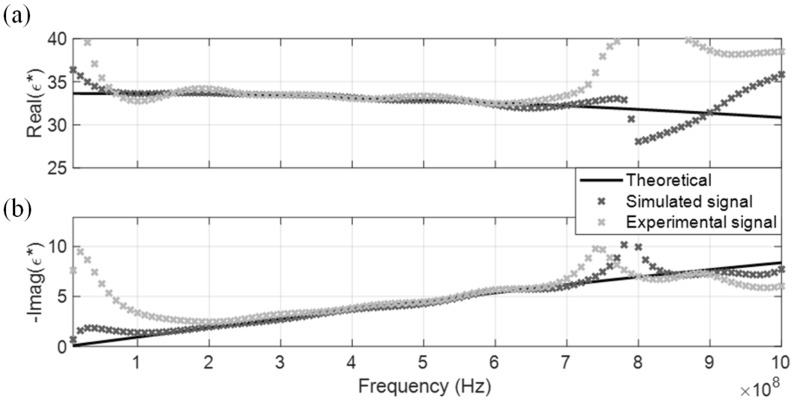
DRA in methanol. (**a**) Real CDP; (**b**) Imaginary CDP.

**Figure 11 sensors-19-01299-f011:**
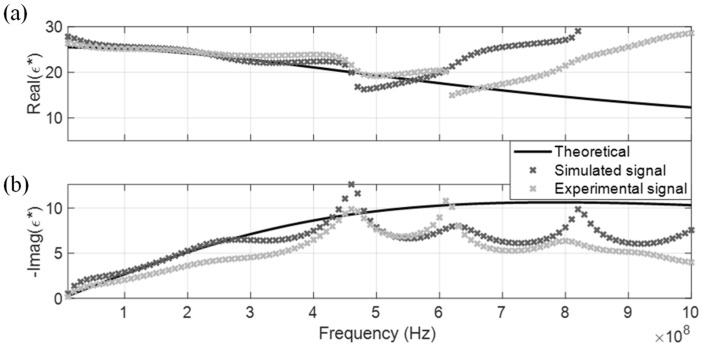
DRA in ethanol. (**a**) Real CDP; (**b**) Imaginary CDP.

**Figure 12 sensors-19-01299-f012:**
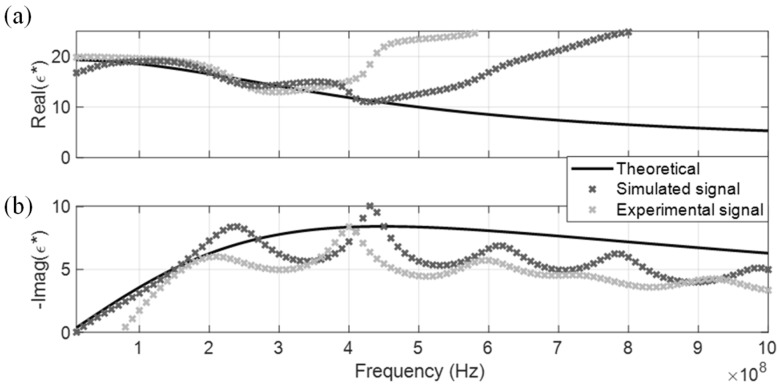
DRA in isopropanol. (**a**) Real CDP; (**b**) Imaginary CDP.

**Figure 13 sensors-19-01299-f013:**
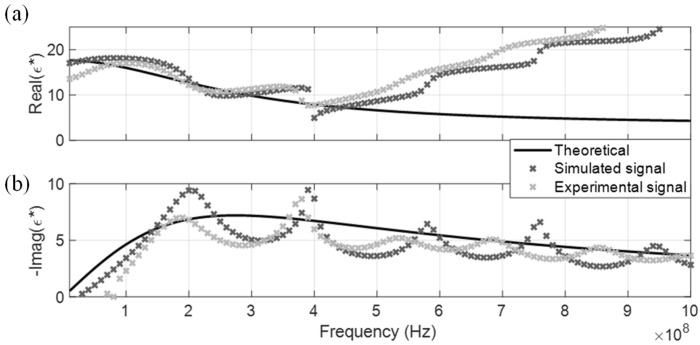
DRA in butanol. (**a**) Real CDP; (**b**) Imaginary CDP.

**Figure 14 sensors-19-01299-f014:**
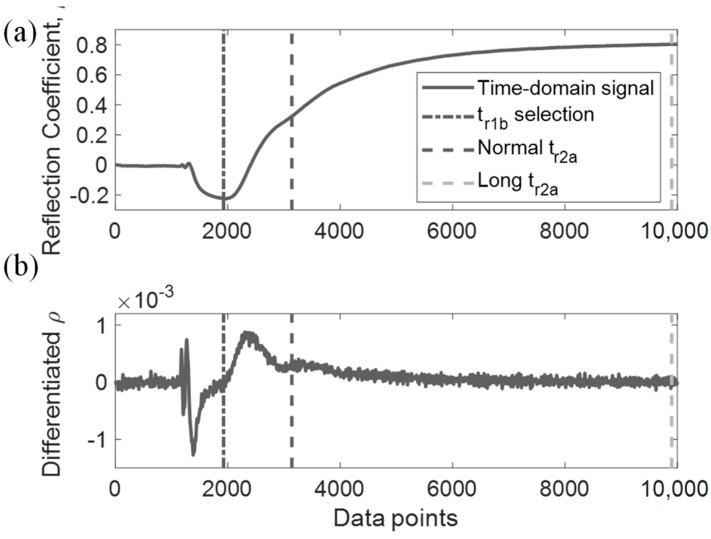
Experimental signal of isopropanol. (**a**) TDR step pulse signal; (**b**) Differentiated signal of (**a**).

**Figure 15 sensors-19-01299-f015:**
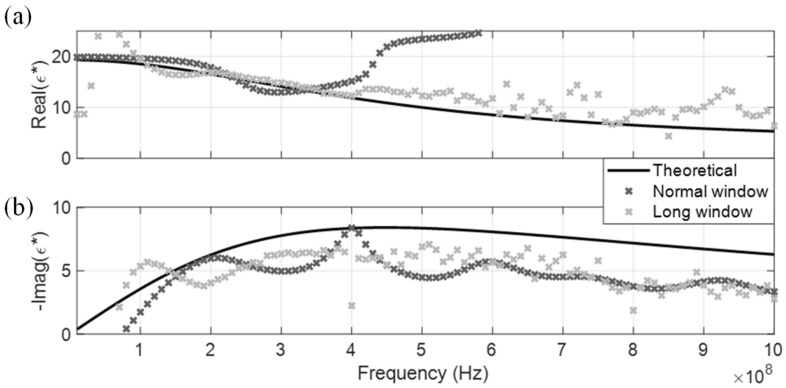
DRA in experimental signal of isopropanol. (**a**) Real CDP; (**b**) Imaginary CDP.

**Table 1 sensors-19-01299-t001:** Cole–Cole parameters of MUT selected for DRA evaluation.

MUT	$\varepsilon_{\infty }$	$\varepsilon_{\mathrm{dc}}$	$f_{rel}$	β	σ
Distilled water [2]	80.20	4.22	17.4 GHz	0.0125	0 μS/cm
Tap water [2]	78.54	4.22	17 GHz	0.0125	300 μS/cm
Acetone [20]	21.20	1.90	47.65 GHz	0	0 μS/cm
Air [16]	1.00	1.00	-	0	0 μS/cm
Methanol [20]	33.64	5.70	3.002 GHz	0	0 μS/cm
Ethanol [21]	25.50	4.25	0.782 GHz	0	0 μS/cm
Isopropanol [14]	19.34	2.48	0.448 GHz	0	0 μS/cm
Butanol [1]	17.70	3.30	0.274 GHz	0	0 μS/cm

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
