# Peer review of "Dielectric Spectroscopy Using Dual Reflection Analysis of TDR Signals"

_sensors, 2019, doi:10.3390/s19061299_

Reviewer 1 Report

The research presents in the paper titled Dielectric Spectroscopy using Dual Reflection Analysis of TDR signal” is very interesting because of the problem of the dielectric constant measure in the materials is very important and difficult.

The authors detail in the introduction the problem and the different proposals that it is possible to use to determine the dielectric constant.

However, in section “2.4 Experimental setup” the authors described the coaxial probe that has been used to measure, but it is missing the equipment employed to do the measurement.

For this reason, it is not clear how they have determined the experimental signal that have been represented in figure 5 to 13.

Moreover, the authors confirm in the “conclusions” section that the DRA of TDR signals methods proposed is the best option to measure the dielectric constant. But there is not any comparison with another method or table in the paper that can confirm it.

Apart from that, I recommend to the authors to check the English. From my point of view, there are some words wrong, like laboratorial…

Finally, I would like to suggest to the authors to separate the system function equation from the figure 1.

Author Response

Response to Reviewer 1 Comments

General comment:

The research presents in the paper titled Dielectric Spectroscopy using Dual Reflection Analysis of TDR signal” is very interesting because of the problem of the dielectric constant measure in the materials is very important and difficult.

The authors detail in the introduction the problem and the different proposals that it is possible to use to determine the dielectric constant.

Response: Thank you for the positive feedbacks and constructive comments. Responses to specific comments are as follows. 

Point 1: However, in section “2.4 Experimental setup” the authors described the coaxial probe that has been used to measure, but it is missing the equipment employed to do the measurement. For this reason, it is not clear how they have determined the experimental signal that have been represented in figure 5 to 13.

Response 1: Details regarding the broadband TDR device and acquisition parameters are included in the revised manuscript. The TDR device is a commercial TDR3000 device from Sympuls Aachen. However, TDR signal acquisition is not limited to the aforementioned device and any readily available TDR device can be used to measure the experimental signals.

Point 2: Moreover, the authors confirm in the “conclusions” section that the DRA of TDR signals methods proposed is the best option to measure the dielectric constant. But there is not any comparison with another method or table in the paper that can confirm it.

Response 2: We stated that “DRA approach is suitable for field applications that can only record limited amount of data points and in-situ dielectric spectroscopy”. We understand that DRA has its own limitation of effective frequency range, especially in dispersive materials. We also mentioned in line 276-278 of Section 3.3 that the multiple reflection analysis (MRA) method perform better characterization of highly dispersive materials, but MRA require much longer time-domain signal to be recorded in order to fully record the steady state.

Other methods to measure the frequency-dependent dielectric constant, such as vector network analyzer (VNA), can measure complex dielectric permittivity (CDP) spectrum better than this approach in laboratory environment. However, we aimed to develop a robust, fast and simple approach to directly perform dielectric spectroscopy in field, even with long cables. To our knowledge, VNA requires long calibration time and demands specifically design sample holders, which renders field dielectric spectroscopy and monitoring extremely challenging. This is discussed in the Introduction section.

Regarding the comparison with other TDR-based approaches, the phase velocity analysis (PVA) method cannot separate the real and imaginary part for comparison. The MRA approach could not be carried out using the acquired waveforms (without reaching the steady state) in this study. Measured values of complex dielectric permittivity that match the theoretical value is the ultimate goal. So the theoretical values are used to evaluate the performance of this approach. As results from other approaches were also commonly presented in comparison with theoretical values, comparison of different approaches can be made from the independent papers accordingly.

Point 3: Apart from that, I recommend to the authors to check the English. From my point of view, there are some words wrong, like laboratorial…

Response 3: We have revised certain tenses and wordings in the abstract according to the suggestions from Reviewer 2. As defined in Oxford Dictionaries, the word “laboratorial” is an adjective which means “devoted to experimental work; characteristic of or appropriate to a laboratory.” While grammatically it’s probably not incorrect, it may be less common. We have changed it to laboratory, using noun as adjective.

Point 4: Finally, I would like to suggest to the authors to separate the system function equation from the figure 1.

Response 4: Thank you for the suggestion. Figure 1 is revised accordingly.

Reviewer 2 Report

The paper "Dielectric Spectroscopy using Dual Reflection Analysis of TDR Signal" presents a method of analyzing the dielectric constant of materials with an analysis of two reflections. The authors claim that this method incurs less callibration effort with reasonable accuracy.

The paper is well organized and well written. The results are clear. The motivation and related work are fine in general terms. The solution is relatively simple to model and it seems hard to believe that this has not been proposed before.

I have the following comments:

- How does the proposed method reduce the callibration efforts? How is this measured or compared to other methods?

- How does the accuracy compare with other methods? Results only show a comparison with theory and with simulation, which are fine. However, I am missing a comparative analysis with other methods (perhaps I got lost in the details).

- I would suggest the authors to provide the figures in pdf or eps to avoid quality loss.

- There is some confusion with the notation (r1 and r2, R1 and R2).

- Abstract: previous contribution should be in present pefect tense: "have been developed", whereas current contributions should be in present tense: "DRA is proposed...", "which requires minimal..."

- What is the difference with the work: "Time Domain Reflectometry Waveform Analysis with Second-Order Bounded Mean Oscillation", Soil Science Society of America Journal 78(4):1146-1152 ?

Author Response

Response to Reviewer 2 Comments

General comment:

The paper "Dielectric Spectroscopy using Dual Reflection Analysis of TDR Signal" presents a method of analyzing the dielectric constant of materials with an analysis of two reflections. The authors claim that this method incurs less calibration effort with reasonable accuracy.

The paper is well organized and well written. The results are clear. The motivation and related work are fine in general terms. The solution is relatively simple to model and it seems hard to believe that this has not been proposed before.

Response: Thank you for the positive feedbacks and constructive comments. Indeed, the solution is relatively simple and it has not been proposed earlier. It’s probably because the reflection ray tracing is usually considered in the time domain and has not been merged to spectral formulation for dielectric spectroscopy. Responses to other specific comments are as follows.

Point 1: I have the following comments:

- How does the proposed method reduce the calibration efforts? How is this measured or compared to other methods?

Response 1: The proposed method only requires two system parameter calibration, namely the probe length and the geometric impedance of the sensing section. The detailed calibration was presented in section 2.2. Compared to other methods that are capable of complex dielectric permittivity (CDP) spectroscopy, system parameters to be calibrated in DRA is way less and the calibration procedure simply requires one known material. The proposed DRA approach only requires system calibration once (after the sensing section is connected), without the need of supplying short, open, load (SOL) condition during the calibration. The broadband TDR reflectometer may also be replaced without recalibrating DRA’s system parameter as the difference of source function is cancelled out in DRA ratio. Other instruments such as vector network analyzer (VNA) requires significantly longer system calibration time and requires calibration whenever the measurement system is moved or restarted.

Point 2: - How does the accuracy compare with other methods? Results only show a comparison with theory and with simulation, which are fine. However, I am missing a comparative analysis with other methods (perhaps I got lost in the details).

Response 2: The accuracy compared to other methods are satisfactory and similar within certain effective frequency range, as discussed in section 3.1 and 3.2. Figures 5-8 and 10-13 demonstrated the comparison of CDP spectrum measured from both simulated and experimental signals to the theoretical CDP spectrum. Direct comparison with other methods were not conducted for reasons. The phase velocity analysis (PVA) method cannot separate the real and imaginary part for comparison. The MRA approach could not be carried out using the acquired waveforms (without reaching the steady state) in this study. Measured values of complex dielectric permittivity that match the theoretical value is the ultimate goal. So the theoretical values are used to evaluate the performance of this approach. As results from other approaches were also commonly presented in comparison with theoretical values, comparison of different approaches can be made from the independent papers accordingly.

Point 3: - I would suggest the authors to provide the figures in pdf or eps to avoid quality loss.

Response 3: Thank you for the reminder. We have provided the figures in separate files in TIFF during manuscript submission.

Point 4: - There is some confusion with the notation (r1 and r2, R1 and R2).

Response 4: The notation was clarified in line 87 of Section 2.1, where the lower case (r1 and r2) and capitalized (R1 and R2) notations represent the time domain and frequency domain variables respectively.

Point 5: - Abstract: previous contribution should be in present perfect tense: "have been developed", whereas current contributions should be in present tense: "DRA is proposed...", "which requires minimal..."

Response : Thank you for the reminder, the tenses are corrected accordingly.

Point 6: - What is the difference with the work: "Time Domain Reflectometry Waveform Analysis with Second-Order Bounded Mean Oscillation", Soil Science Society of America Journal 78(4):1146-1152 ?

Response : DRA measures the CDP spectrum with frequency information, in which the real and imaginary parts of CDP are decoupled. The mentioned work focused on the time-domain approach to determine the travel time from TDR signals and calculate the single-valued apparent dielectric constant (K_a) without any frequency information.

Round  2

Reviewer 1 Report

The article entitled "Dielectric Spectroscopy using Dual Reflection Analysis of TDR Signal" has been carefully modified and well revised.

The work is supposed to be finally accepted for publication in Sensors.